



# Closing the phenotyping gap with non-invasive belowground field phenotyping

Guillaume Blanchy[1], Waldo Deroo[2], Tom De Swaef[3], Peter Lootens[4], Paul Quataert[5], Isabel Roldán-Ruíz[6], and Sarah Garré[7]

[1]University of Liège, Belgium, Flanders Research Institute for Agriculture, Fisheries and Food (ILVO), Ghent University (UGent), F.R.S.-FNRS (Fonds de la Recherche Scientifique)
[2]Flanders Research Institute for Agriculture, Fisheries and Food (ILVO)
[3]Flanders Research Institute for Agriculture, Fisheries and Food (ILVO)
[4]Flanders Research Institute for Agriculture, Fisheries and Food (ILVO)
[5]Flanders Research Institute for Agriculture, Fisheries and Food (ILVO)
[6]Flanders Research Institute for Agriculture, Fisheries and Food (ILVO)
[7]Flanders Research Institute for Agriculture, Fisheries and Food (ILVO)

*Corresponding author: Sarah Garré (sarah.garre@ilvo.vlaanderen.be)

**Abstract.** Breeding climate-robust crops is one of the needed pathways for adaptation to the changing climate. To speed up the breeding process, it is important to understand how plants react to extreme weather events such as drought or waterlogging in their production environment, i.e. under field conditions in real soils. Whereas a number of techniques exist for above-ground field phenotyping, simultaneous non-invasive belowground phenotyping remains difficult. In this paper, we present the first dataset of the new HYDRAS open access field phenotyping infrastructure, bringing electrical resistivity tomography, alongside drone imagery and environmental monitoring, to a technology readiness level closer to what breeders and researchers need. This paper investigates whether electrical resistivity tomography (ERT) provides sufficient precision and accuracy to distinguish between belowground plant traits of different genotypes of the same crop species. The proof-of-concept experiment was conducted in 2023 with three distinct soybean genotypes known for their contrasting reactions to drought stress. We illustrate how this new infrastructure addresses the issues of depth resolution, automated data processing, and phenotyping indicator extraction. The work shows that electrical resistivity tomography is ready to complement drone-based field phenotyping techniques to accomplish whole plant high-throughput field phenotyping.

## 1 Introduction

Alongside actions to mitigate climate change, the agricultural sector needs solutions to adapt to the increased occurrence of weather extremes such as drought or waterlogging. In this sense, breeding climate-robust crops is one of the needed pathways for adaptation to climate change (Snowdon et al., 2020). In a typical breeding program, the selection of a new variety ready for the market takes more than a decade (Voss-Fels et al., 2019). To speed up this process and breed climate-robust crops more efficiently, it is important to understand how plants respond to extreme weather events such as drought or waterlogging, and to



identify which traits should be targeted in selection programs. Recent advances in phenotyping have resulted in powerful tools
to screen plant traits in large collections of plants in different settings and under various conditions.

While methods for evaluation under controlled conditions in growth chambers or greenhouses remain important in plant
phenotyping, a weak correlation has often been reported between responses in a controlled environment and a production
environment (Langstroff et al., 2021). Consequently, on-field evaluation in combination with remote sensing techniques

is increasingly being deployed (Araus and Cairns, 2014). Nevertheless, few options are currently available to phenotype
belowground in field conditions in undisturbed and living soils. Most techniques for soil-root investigation in the field are
invasive and destructive (Das et al., 2015; Trachsel et al., 2010). These techniques cannot monitor a given plant, or plot,
throughout its entire growth period. Installation of minirhizotron tubes with cameras offers a less invasive approach (Svane et al.,
2019), but delivers only local information on the (partly disturbed) zone surrounding the tubes (Rajurkar et al., 2022; Vamerali

et al., 2011). Electrical Resistivity Tomography (ERT) is a technique injecting current in a pair of electrodes and measuring the
voltage in another pair. By repeating this procedure over many combinations of electrodes along a transect, one can reconstruct
the distribution of electrical resistivity in the subsurface below the transect. ERT integrates the entire soil volume under a plant,
row, or plot and is sensitive to changes in soil moisture. This makes the technique suitable to monitor the impact of crop root
system on soil water depletion, which is related to static or dynamic root system traits (McGrail et al., 2020; Atkinson et al.,

2019; Ehosioke et al., 2020). Wasson et al. (2020) highlighted the potential of non-invasive root phenotyping techniques and
indicated several advantages. The most important advantage is that ERT does not disturb root system structures, architecture, and
functions within the rhizosphere and bulk soil environments, so that the rhizosphere can be monitored as a 'holistic phenotype'.
This could lead as far as the discovery of new traits to be targetted by breeders and is essential for researchers investigating the
functioning of the soil-plant system.

ERT (also denoted as Electrical resistivity imaging (ERI) or direct current resistivity (DCR)) has been used before to assess
static and dynamics properties of the root zone in the context of agronomical, ecological and engineering studies. In the
agronomic context, Michot et al. (2003) were amongst the first to use ERT to observe root water uptake patterns of maize in the
field. In the following years, the potential of ERT to monitor soil water depletion in the root zone was further demonstrated
by various authors on a range of crops, soils, and climates (amongst others by Amato et al. (2009), Srayeddin and Doussan

(2009), Cassiani et al. (2012), Garré et al. (2013), Blanchy et al. (2020c), . . . ). Whalley et al. (2017) were the first to bring the
technique explicitly to the breeding context by testing it to discriminate the soil moisture profile under 13 wheat varieties during
3 years. Amongst the tested techniques, ERT provided the best discrimination among wheat lines. They found inter-genotype
differences in depth of water uptake and in the extent of surface drying, paving the way for ERT as a technique for belowground
plant phenotyping complementing aboveground High-Throughput Field Phenotyping (HTFP).

Despite its clear potential, past studies highlighted a few remaining challenges in using ERT as a fully-fledged belowground
phenotyping technique. In agronomic applications, ERT is mostly implemented as a surface transect or grid. To maximise
resolution in the root zone, small electrode spacings (0.2-0.5 m) should be adopted. Nevertheless, the resolution and sensitivity
of the imaging declines with depth. Obtaining high spatial resolution throughout the root zone while attaining sufficient depth
penetration is a significant challenge (Zhao et al., 2019). The resolution of ERT is typically limited to the decimeter range





in the field, especially when targeting (the effects of) roots at greater depths. The effect of small uncertainties in electrode positions quickly inflates when electrode spacings are reduced and when surface electrodes are combined with buried or borehole electrodes. In addition, the choice of electrode configuration significantly influences the quality of ERT data and the sensitivity distribution (Garre et al., 2021). Optimizing electrode layouts and measurement arrays for specific phenotyping objectives (balancing space and time resolution) is therefore key to get robust and accurate phenotyping datasets (Uhlemann et al., 2018).

Another challenge is that apparent resistivity measurements need to be converted to resistivity distributions by an inversion process if spatially explicit information is required. The classical geophysical inversion is ill-posed and choosing the most appropriate inversion parameters still requires significant expert knowledge. In addition, the quality of the inversion results is sensitive to several factors, which may sometimes change during the growing season (e.g. changing contact resistances due to drought). Developing robust inversion algorithms and improving the accuracy of the inversion process is necessary to

develop standardised processing pipelines and the resulting indicators for breeders. This might come from different types of inversion strategies, such as joint or coupled inversion, machine learning, or other types of hybrid algorithms. As a geophysical inversion is ill-posed, ERT data inversion results are inevitably associated with uncertainties. Understanding and quantifying these uncertainties (Tso et al., 2021; Linde et al., 2017) and developing strategies to manage and communicate them are important for the reliable interpretation of resistivity distributions and derived phenotyping indicators.

Finally, establishing standardized (meta-)data formats, procedures for ERT measurement for phenotyping and calibration and data processing is crucial for ensuring consistency, comparability, and interoperability of data across different studies and locations. Currently, typically very little information about the exact settings of the measurement device, electrode array and data processing is shared in research publications. Some initiatives, such as REDA try to address this, but are currently not widely adopted. Well-documented, open access ERT datasets for agriculture, and especially phenotyping are rare (CAGS).

There are some efforts for unified data models in the phenotyping community (e.g. MIAPPE), environmental monitoring (e.g. eLTER, ENVRI-FAIR, Wohner et al. (2022)) and in geoscience (e.g. ODM2, ODMX), but those remain limited to certain subcommunities or projects. In addition, there is no clear, broadly accepted standard for the ERT data storage, processing and sharing in the framework of belowground phenotyping. This hampers the wide use and application of the technique and/or resulting data sets.

HYDRAS (HYdrology Drones and RAinout Shelter) is an open access field phenotyping infrastructure, located nearby Ghent in Belgium. It was designed to address these challenges and bring electrical resistivity tomography, alongside drone imagery and environmental monitoring, to a technology readiness level closer to what breeders and researchers of the soil-plant-atmosphere system need. The objective of this work was to investigate whether electrical resistivity tomography (ERT) provides sufficient precision and accuracy to distinguish between belowground plant traits of different genotypes of the same crop species. We

address the issues of depth resolution, automated data processing, phenotyping indicator extraction and open, interoperable data sets. Using the data of a proof-of-concept (POC) experiment conducted in 2023, we illustrate steps towards addressing these challenges and highlight potential further developments.



## 2  Methodology

The HYDRAS open field phenotyping infrastructure comprises fields with mobile rain-out shelters to generate precise drought periods and neighbouring control fields. Regular drone flights with RGB and multispectral cameras generate high-throughput phenotyping indicators characterizing the above-ground part of the plant at key phenological stages. Continuous electrical resistivity tomography (ERT) monitoring provides indicators representing the plant below the ground. Measurement methods are calibrated and validated with independent soil and plant sensors. In this work, we present the results of the belowground phenotyping activities during the proof-of-concept (POC) experiment conducted in 2023 with three distinct soybean genotypes known for their contrasting reactions to drought stress: *Glycine max* (L.) Merr. cv. Lenka, *Glycine max* (L.) Merr. cv. Hermes and *Glycine max* (L.) Merr. cv. Pro-1. The Lenka genotype is known to be more resistant to drought thanks to the slow canopy drought trait (Ye et al., 2019). Pro-1 en Hermes do not have this trait, but the Pro-1 genotype has a more compact *habitus* that can potentially make it more drought tolerant.

### 2.1  Site description

HYDRAS is located in Melle, Belgium (50.99281N, 3.78602E) on a sandy soil classified as a *Eutric Stagnic Glossic Retisol* according the WRB (2022) or *Sbc* according to the Belgian soil classification system. Each drought and control block consists of three fields undergoing a 3-year crop rotation, with 6 blocks in total (see Figure 1). The size of each field is 30 m x 10 m. The drought blocks have an additional parking for the shelters, which is not used for experiments. 2023 was the first year of operation of the infrastructure. Two weather stations are present: one underneath a shelter and one in open air. HYDRAS is equipped with a calibration pit. In this pit, soil moisture, soil temperature and electrical resistivity sensors monitor the soil status continuously in each horizon (see below for details). This allows us to establish a robust field-specific pedophysical relationship and provide continuous validation data characterizing the field status. In addition, soil moisture and water potential sensors are installed at the soil surface in the experimental fields to validate the ERT measurements during the growing season.

### 2.2  Data acquisition

#### 2.2.1  Electrical resistivity tomography

In HYDRAS, Electrical Resistivity Tomography (ERT) is used to identify differences in root system functioning of a panel of plant genotypes by monitoring the impact of each genotype on soil water depletion patterns. Dryer soil, where roots are extracting water, has a larger resistivity (lower electrical conductivity) than wetter soil around the root zone. ERT quantifies the bulk electrical resistivity of the soil between a series of electrodes. The bulk electrical resistivity corresponds to the combined resistivity of soil particles, pore water and air. A basic measurement system consists of four electrodes (A, B, M, N), often referred to as a "quadrupole". A direct current of known intensity ($I$) is sent between electrodes A and B, while a potential difference ($V$) is measured between electrodes M and N. The resistivity meter switches the polarity of the current using a square wave to avoid polarisation of the injection electrodes. It then computes a transfer resistance ($R$, Ohm) for each combination of



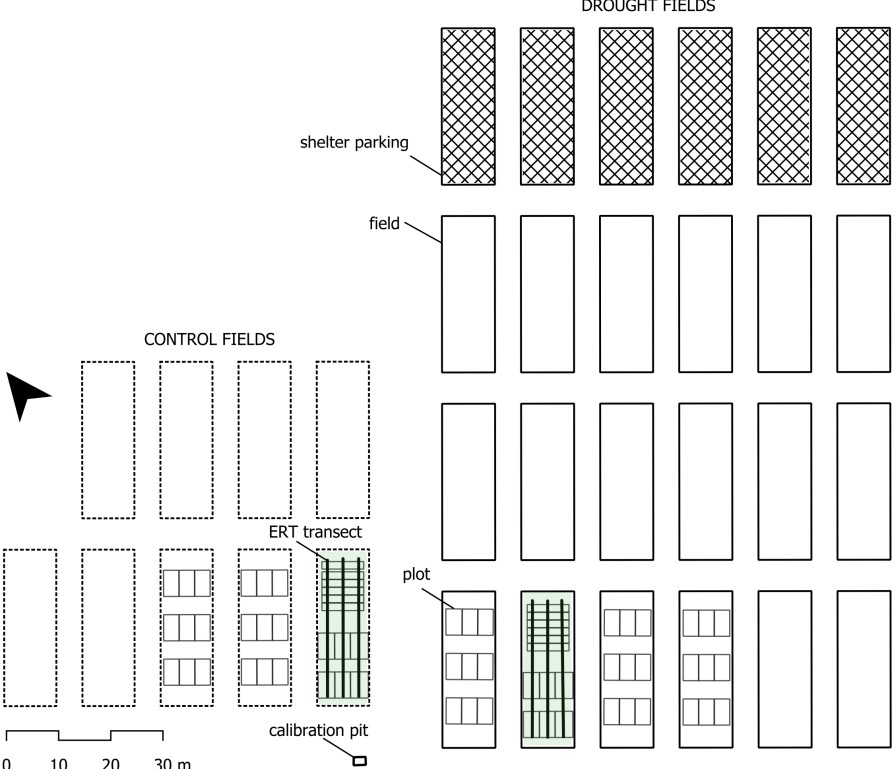

**Figure 1.** Lay-out of the HYDRAS field phenotyping infrastructure during the POC2023 experiment. The control and drought field equipped with ERT in 2023 are indicated in shaded green. Black lines indicate the location of an ERT transect constituted of surface electrodes and electrodes buried at 0.5 m depth. The polygons inside the fields represent the plots sown with soybean.

four electrodes, based on Ohm's law $R = V/I$. Based on the distances between the four electrodes, a geometric factor $K$ can

be computed to transform this transfer resistance into a bulk apparent electrical resistivity ($\rho_a$, Ohm.m): $\rho_a = K \times R$. This resistivity is "apparent" because it represents the resistivity of a homogeneous isotropic ground with the same transfer resistance. Electrical conductivity (EC, mS/m) is the inverse of the resistivity.

During the POC experiment, we equipped one field in the control block and one in the drought block with three ERT transects crossing the soybean genotype plots (the green shaded fields in Figure **??**). Each transect was 26.4 m long (excluding the borders

of the field) and consisted of a surface cable and a cable buried at 0.50 m depth. The buried cable increases the resolution in the bottom part of the root zone. The surface electrode spacing was 0.30 m, whereas the electrode spacing along the buried cable was 0.60 m (see Figure 2a and b). This resulted in a total of 426 electrodes per field connected to the ERT base unit. The surface electrodes had a diameter of 0.01 m and a length of 0.1 m and were connected to the multicore cables. The cable take-outs of the buried cables served directly as electrodes. In the HYDRAS installation, the buried cables stay in place permanently, since they



are installed under the plough depth. The surface cables are put in place after sowing, stay there the entire growth period and are
removed at harvest.

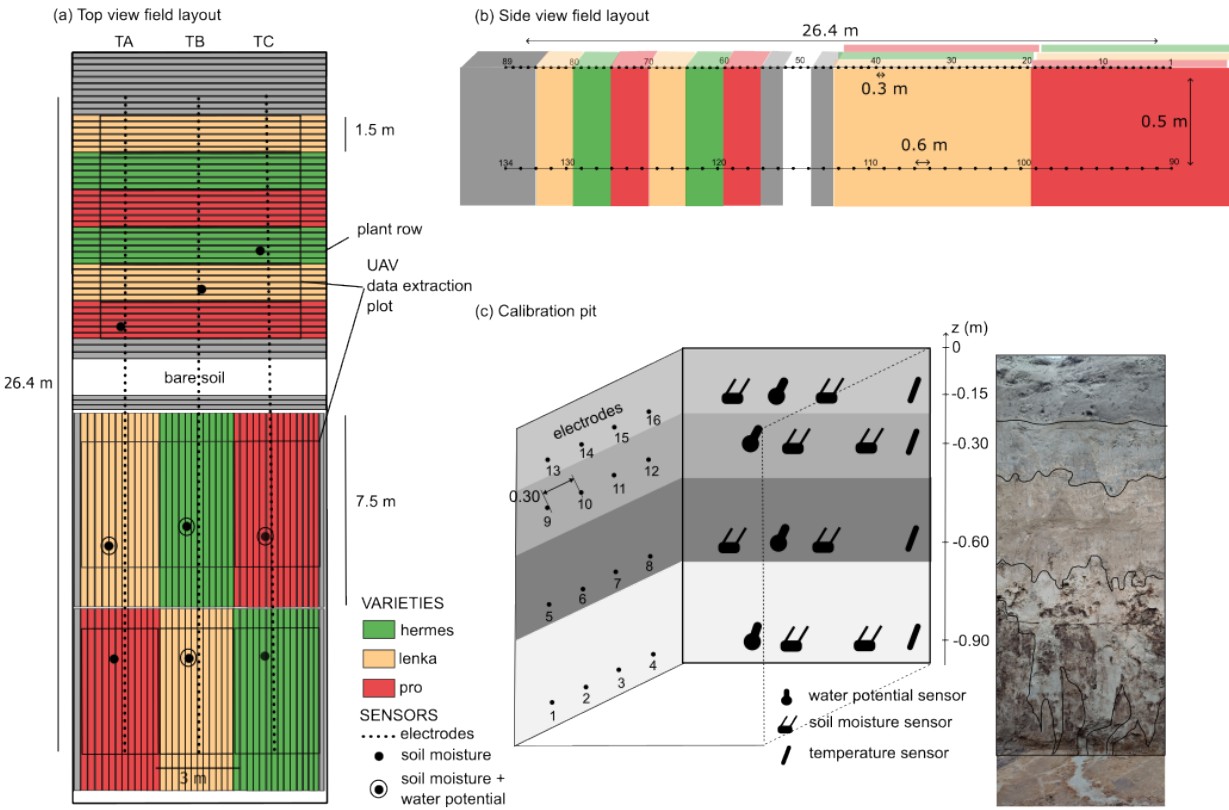

**Figure 2.** a) Top view scheme of the ERT equipped fields in HYDRAS during the POC2023 experiment. Colours represent genotypes. Small
dots represent electrodes of the three ERT transects: TA, TB and TC; (b) Side view scheme of the ERT equipped fields; (c) Calibration pit with
sensor, electrode locations and soil profile.

We performed the ERT measurements with a custom-made, single channel, multi-electrode resistivity meter developed
by Subsurface Insights (Hannover, NH, United States) and Metinco (Wapenveld, The Netherlands). One system can support
thousands of electrodes in groups of 32. The system is self-contained: it has an internal single board computer which controls the
data acquisition and communicates with a cloud server for data storage and further processing. The system can inject currents
up to 12 mA or, alternatively, apply a fixed transmitter (*Tx*, between electrodes A and B) voltage between 0 and 150 V. In
the POC experiment, we applied a Tx voltage of 20 V. This Tx voltage was found adequate to have a good signal-to-noise
ratio for our electrode spacing and soil type. The measurement sequence for the POC experiment contained dipole-dipole and





gradient quadrupoles on the same line and between surface and buried lines (full sequence available as appendix). We collected
reciprocals of all quadrupoles for error assessment. One ERT measurement sequence of 2118 quadrupoles took about 1.5 h and
was repeated continuously from 2023-06-21 (just after sowing) to 2023-09-30 (just before harvest).

As Figure 2a shows, we adopted two types of plant row orientation in this POC experiment (along and perpendicular to the
ERT transects) to assess whether this orientation influences the measurements. The orientation did not affect the ERT-derived
indicators considerably, confirming that the 2.5D assumption (homogeneity in the Y direction) holds for both row orientations in
this trial. Analysis of the yield data of the trial revealed that the yield in the plots with rows perpendicular to the ERT transects
was systematically lower than in the plots with rows established along the ERT transects. This was most likely due to more edge
effects and impact of tractor wheel tracks (data not shown).

### 2.2.2  Environmental monitoring

Four soil water potential (TEROS21, Meter group) and nine soil moisture sensors (CS616, Campbell scientific) were installed
vertically in the soil surface. Figure 2a shows the location of those sensors in the field. They are all maximum two plant rows
away from the ERT transects. The site has a calibration pit equipped with soil moisture, soil water potential, soil temperature
sensors (T107, Campbell Scientific) and four electrodes in each of the four soil horizons up till 1 m depth (see Figure 2c).
Sensors and electrical resistivity measurements are continuously logged throughout the year to establish the field-scale soil
hydraulic functions, thermal characteristics and pedophysical calibration functions. Figure 3b shows significant variability near
the soil surface, but at 0.6 and 0.9 m depth the soil moisture remained rather constant throughout the season.

Meteorological variables (solar radiation, precipitation, wind direction, wind speed, air temperature, water vapour pressure,
atmospheric pressure) were measured using a ATMOS41 weather station (Meter group). One station was mounted in open air
and the other under the rainout shelter when drought was applied to assess the impact of the shelter on the microclimate.

### 2.2.3  Above-ground phenotyping using drone imaging

Drone-based high-throughput field phenotyping was executed in analogy with Borra-Serrano et al. (2020) and Pranga et al.
(2021). In total 14 flights were performed using a UAV DJI Matrice 600 Pro (DJI, Shenzhen, China) equipped with an RGB
camera (a6400, Sony Corporation, Tokyo, Japan), a 10-band multispectral camera (M Dual Camera System, MicaSense, Seattle,
USA), and a thermal camera (WIRIS Pro, Workswell, Czech Republic). For this manuscript, only multispectral data was used.
Flight speed and flight altitude were 2.7 m/s and 30 m for the multispectral sensor. This resulted in a ground sampling distance
of 2.0 cm/pixel. All flights were conducted within 2 h of solar noon and with 80-80% side and front overlap. Multispectral
images were processed and geo-referenced orthophotos were created using PIX4Dmapper 4.7.5 (Pix4D S.A., Switzerland).
Several plant indices were calculated, but here we only show the Soil-Adjusted Vegetation Index (SAVI). This index minimizes
the influence of soil brightness using a correction factor. NIR represents the pixel values from the near infrared band, Red the
pixel values from the near red band and L the amount of green vegetation cover.







**Figure 3.** Overview of environmental conditions in the calibration pit, drought, and control fields. (a) Precipitation and cumulative precipitation deficit (= $\Sigma(ET_0 - P)$) under drought and control treatments. The cyan irrigation event was only applied on the control plots. The yellow band indicates the length of the applied drought treatment with the rainout shelter. (b) Soil moisture in the calibration pit at four depths (grey hues) and in the drought (orange hues) and control fields (blue hues) at 0.15 m depths (different hues = different locations in the field). (c) Soil water potential in the calibration pit at 4 depths (grey hues) and the drought (0-0.10 m depth) and control fields (vertical installation depth: 0-0.10 m). (d) Soil temperature in the calibration pit at 4 depths (grey hues) and in the control treatment (vertical installation depth: 0-0.10 m).





$$SAVI = \frac{(NIR - Red)}{(NIR + Red + L)} * (1 + L) \tag{1}$$

Generally, in areas with no green vegetation cover, L=1; in areas of moderate green vegetative cover, L=0.5; and in areas with very high vegetation cover, L=0 (which is equivalent to the NDVI method). This index outputs values between -1.0 and 1.0.

### 2.3 Processing and quality control

Figure 4 shows an overview of the data processing workflow. The workflow is divided into four different steps: preprocessing, quality assessment, inversion and computing indicators. More explanations on some of these steps are given below.

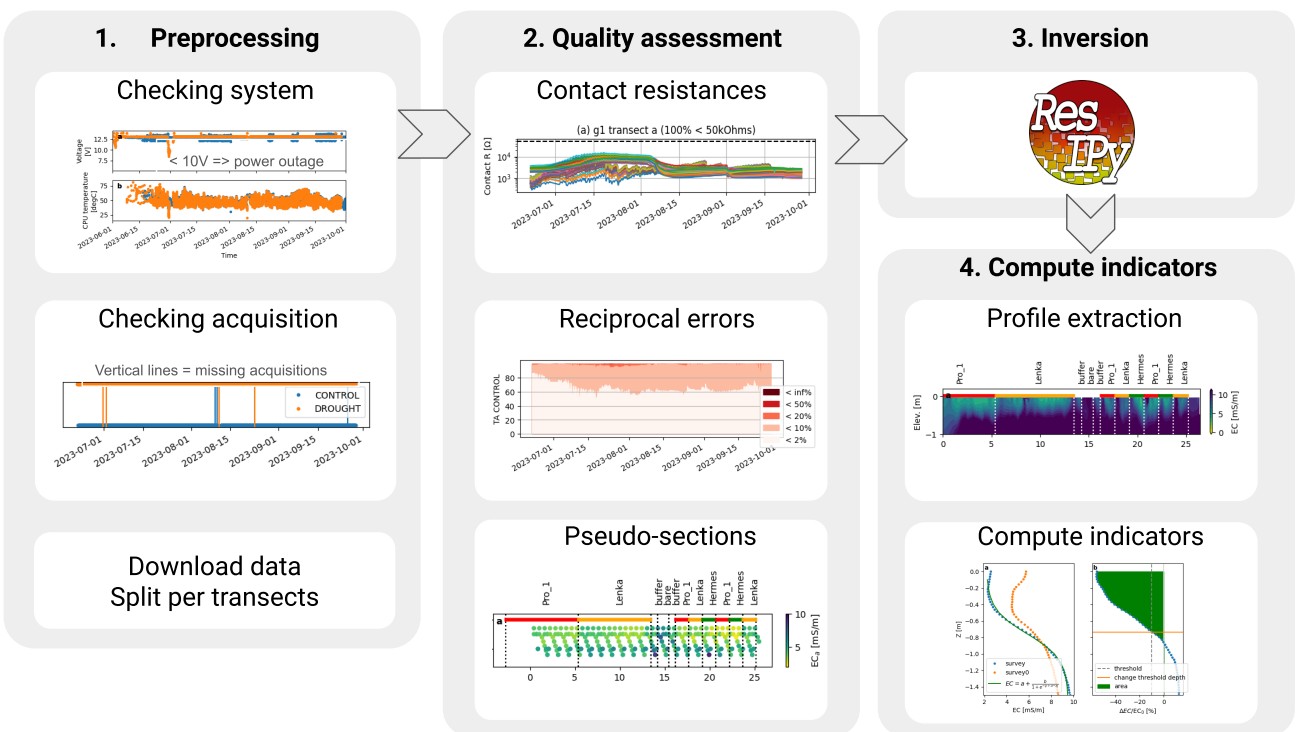

**Figure 4.** Overview of the HYDRAS below-ground phenotyping data processing pipeline

### 2.3.1 Quality assessment

From the field ERT unit, the data is continuously uploaded to a server, where a basic quality assessment at the level of the quadrupoles is executed. Users are alerted when the server does not receive data or for defined thresholds in raw data (e.g. very low injection current). Figure 5 shows the evolution of the raw metrics for each measured quadrupole over the entire monitoring period: current (I), measured voltage ($V_{mn}$), contact resistance (cR, estimated resistance between the electrode and the soil),



apparent resistivity ($\rho_a$), standard deviation from stacking (dev) and relative difference in apparent resistivity to the reference background image ($\Delta\rho_a/\rho_{a0}$) from the first survey ($\rho_{a0}$). This overview enables us to spot any irregularities in the system or sudden environmental changes, such as heavy rainfall after a dry period (e.g. end of August), allowing for real-time alerting during the monitoring period.

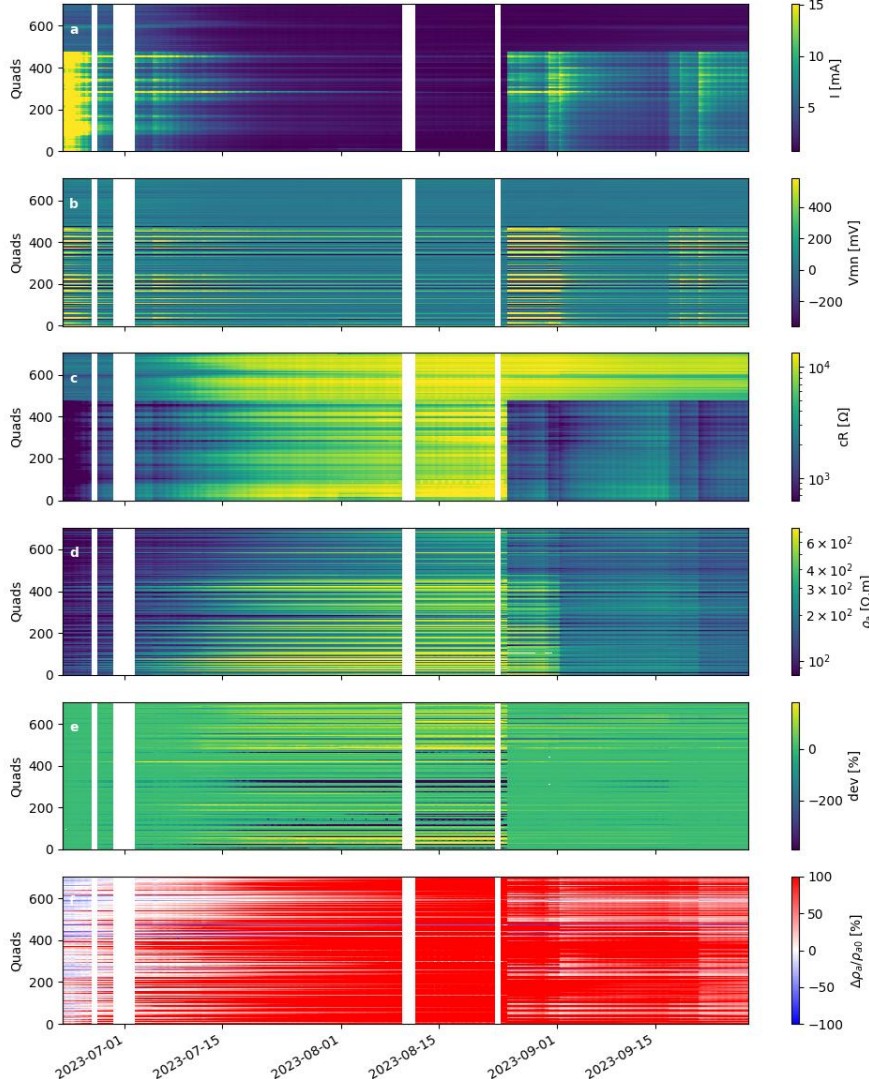

**Figure 5.** *Evolution of raw metrics at quadrupole level (Quads) measured by the ERT system featuring injected current (I), measured voltage (Vmn), contact resistances (cR), apparent resistivity ($\rho_a$), standard stacking deviation (dev), relative change in apparent resistivity from the first survey ($\Delta\rho_a/\rho_{a0}$). White vertical bands show no-data time. The data are shown here for the transect A of the ERT field on which the drought treatment was applied (see Figure 2 for transect location within the field).*





Quadrupoles with electrodes often associated with high contact resistances (>50 kOhms) were removed from the datasets. Negative apparent resistivity as well as quadrupoles with large (>50%) standard deviation were also removed before further processing. In addition, quadrupoles with injection electrode pair buried and voltage electrode pair at the surface (or the opposite) were removed from the dataset. After preliminary inspection of the raw data, we noticed that these quadrupoles lead to significant artefacts around the electrodes, possibly caused by small inaccuracies in the position of the surface cable with respect to the

buried cable. From synthetic studies, we found that 2 cm misplacement of electrodes (longitudinal or lateral to the transect) can reproduce the artefacts we observed if all data are used to invert the dataset (data not shown). Given the relatively small electrode spacing of the survey and the difficulty to position the surface electrode right on top of the buried ones with centimetric precision, we decided to remove these quadrupoles from the inversion.

The pipeline computes reciprocal errors and Figure 6 shows the evolution of their distribution as a function of time for each

transect. The reciprocal errors tend to increase as the soil dries out, because this also increases the contact resistance of the electrodes at the surface (see Figure 5). The reciprocal errors were not used for filtering, but as a weight during the inversion. We fitted a power-law error model for each survey on a transect using the binned reciprocal errors (Koestel et al., 2008). This approach ensures a sufficient number of data points to obtain a robust error model, while allowing the error model to vary throughout the season. Further details on the data processing can be found in the accompanying Jupyter notebook.

### 2.3.2 Data inversion

The ERT data were processed using the ResIPy software v3.5.1 (Blanchy et al., 2020a) that makes use of the Occam's based R2 inversion code (Binley, 2015). Each survey was inverted and compared to the first recorded survey following the difference inversion method (LaBrecque and Yang, 2001) (reg_mode = 2 in ResIPy). The difference inversion consists in first inverting a reference survey (in our case, the first survey collected on 2023-06-21), and then computing, for each subsequent survey, the

difference with respect to this reference survey. For a given survey, the response (i.e. transfer resistances for each quadrupole) from the inverted model of the reference survey is computed and added to the difference between the measured transfer resistances of the reference and the given survey. This new dataset of transfer resistances is then inverted. This approach highlights differences between survey dates and suppresses the systematic errors that might arise due to electrode placement. It is well suited when electrodes are kept in place between surveys, as was the case in this study. The inversion procedure was done

using a triangular mesh. The objective function to be minimized consisted of a data misfit (weighted by the errors computed from the fitted error model) and model misfit (smooth L2 regularization). Inversions typically converged within 5 iterations and reached a final weighted root-mean-square error close to 1. More details on the inversion can be found in Binley and Slater (2020). Appendix C shows several of these inversion quality indicators as a function of time for all transects.

### 2.3.3 Temperature correction and conversion to soil moisture

After inversion, resistivity data were temperature-corrected and converted to soil moisture using a site-specific pedophysical relationship, established using multi-sensor data in the HYDRAS calibration pit (see Figure 3d). The resistivity data of this study are corrected for temperature using the following relationship (Ma et al., 2010; Campbell et al., 1949) with $\alpha = 0.02$. The





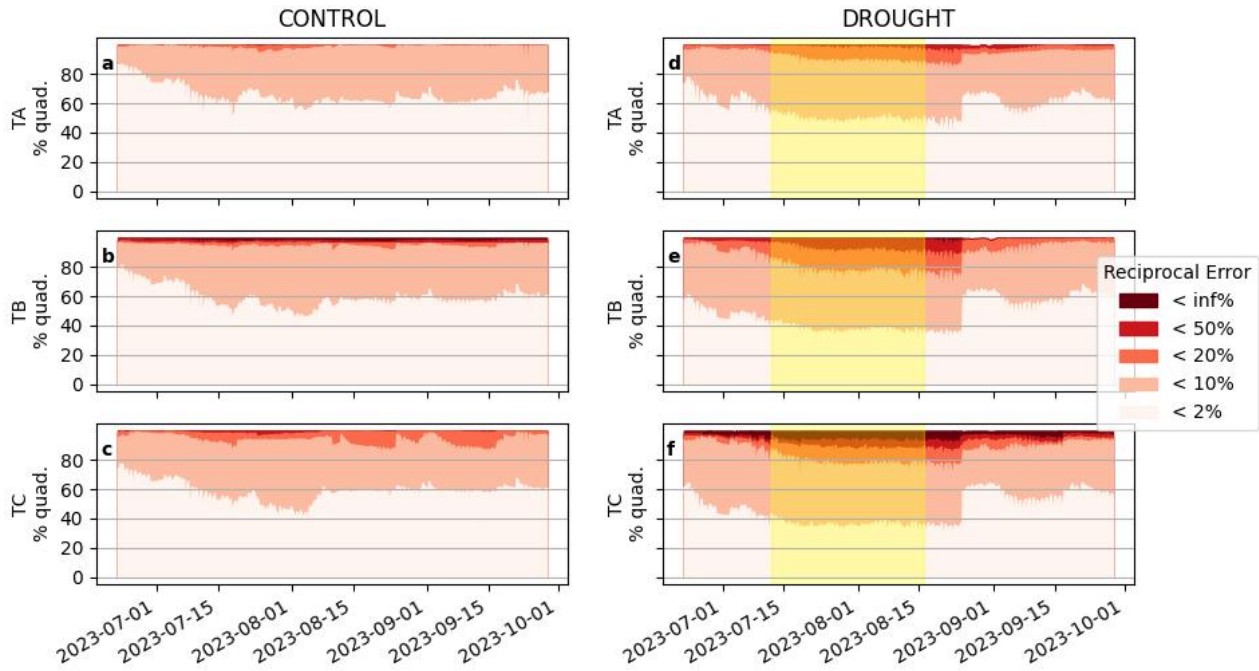

**Figure 6.** *Evolution of the distribution of reciprocal errors ($\frac{|R_N - R_R|}{\frac{R_N + R_R}{2}} \times 100$) during the growing season of the POC23 experiment for all ERT transects (TA, TB and TC) in CONTROL (subplots a,b,c) and DROUGHT (subplots d,e,f). Reciprocal errors are distributed in classes for easier visualisation. The percentage of quadrupoles in each class is indicated on the vertical axis.*

soil temperature profile during the measurements was assumed to be equal to the temperature measured in the different soil horizons of the calibration pit, which is a simplification of reality.

$$EC_{b,20} = \frac{EC_b}{[1 + \alpha(T - 20)]} \qquad (2)$$


Figure 7 shows the sensor data from the calibration pit for the four soil horizons. As the deepest layers never reached dry conditions, the in-field pedophysical relationship for those horizons cannot be established in the dry range at the moment. Further monitoring in coming years will improve the amount of data in the dryer range. For illustration, we fit the simplified Waxman and Smits model to the data of each soil horizon.





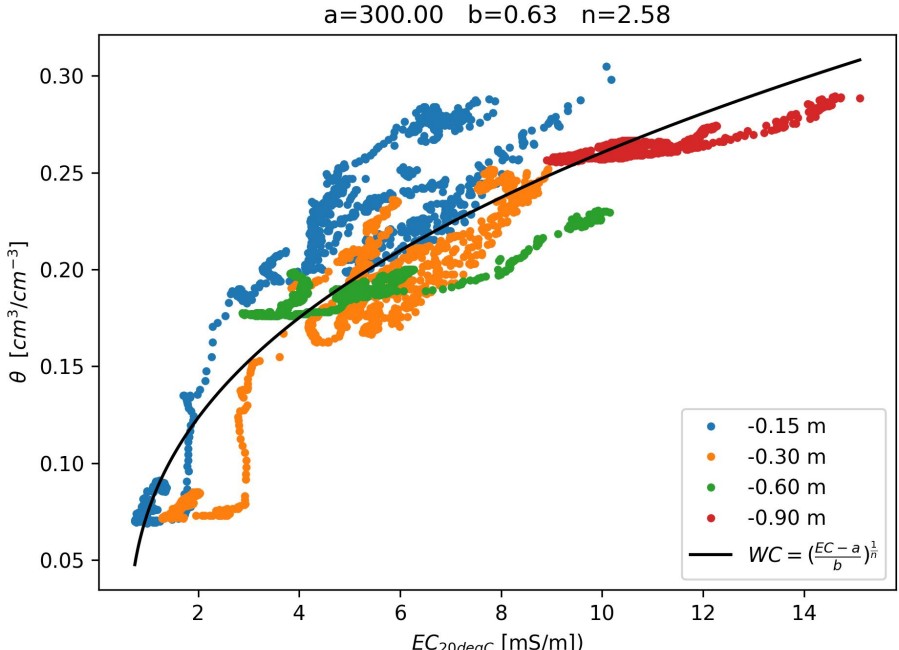

**Figure 7.** Evolution of volumetric soil moisture $\theta$ with respect to the bulk temperature-corrected electrical conductivity ($EC_{b,20}$) based on the data collected at the different depth in the control pit. For illustration, the simplified Waxman and Smits model was fit to the data of each soil horizon as shown in Garré et al. (2013) (black line on top of the data points).

### 2.3.4 Data dissemination: from geophysicists to plant scientists and breeders

While profiles of inverted electrical conductivity are certainly interesting to geophysicists, they have less meaning for plant scientists or breeders who use the HYDRAS open access infrastructure. As such, we translated the geophysical results into phenotyping indicators that are more interpretable for crop breeders and researchers. Table 1 shows four of those indicators with potential. We computed indicators along the inverted profile and the profile of difference compared to background. Figure 8 illustrates how these indicators are computed in $EC$ or $\Delta EC$ profiles. The shape of the inverted EC profile often takes the form of a sigmoid (Shanahan et al., 2015). The parameters of the sigmoid summarize information about the shape of the soil moisture profile. Fitting parameters a defines an offset from 0 soil moisture and b controls the width of the sigmoid (magnitude of the drying), $d$ controls the steepness of the curve, and $c$ defines the depth of inflection in the soil profile. Parameters $c$ and $d$ influence the drying depth. The integration of the negative difference in EC with depth (green shaded zone in Figure 8b) gives an estimation of the extent of the water depletion caused primarily by root water uptake. Similarly, the percentile 10% of the depths at which a negative difference larger than a threshold of 20% change occurs (dashed vertical line in Figure 8b) is an indicator for the advancement of the drying depth in the soil profile (orange horizontal line). The pedophysical relationship shown above allows to translate EC to soil moisture (not done here).



**Table 1.** Description of the proposed phenotyping indicators derived from the average electrical conductivity profiles.

| Indicator | Meaning | Formula |
|---|---|---|
| Fitting parameters sigmoid (subplot a) | Shape of the soil moisture profile at a specific date | $EC = a + \dfrac{b}{1 + e^{-(c + d*z)}}$ |
| $a,b$ | *offset & asymptote of the sigmoid, control magnitude of the drying* | |
| $c$ | *depth of inflection of the profile* | |
| $d$ | *steepness of the sigmoid around the inflection point* | |
| Drying area (DA) (green area subplot b) | Integrative total water uptake from the soil profile since the start of the experiment and a specific date | $\int_0^{1.5} \frac{\Delta EC}{EC_0} dz$ where $\frac{\Delta EC}{EC_0} < 0\%$ |
| Drying depth (DD) (orange line subplot b) | Percentile 20% of all depths in the soil profile where a -20% change as compared to the first survey has occurred at a specific date. | $q10(z)$ where $\frac{\Delta EC}{EC_0} < -20\%$ |

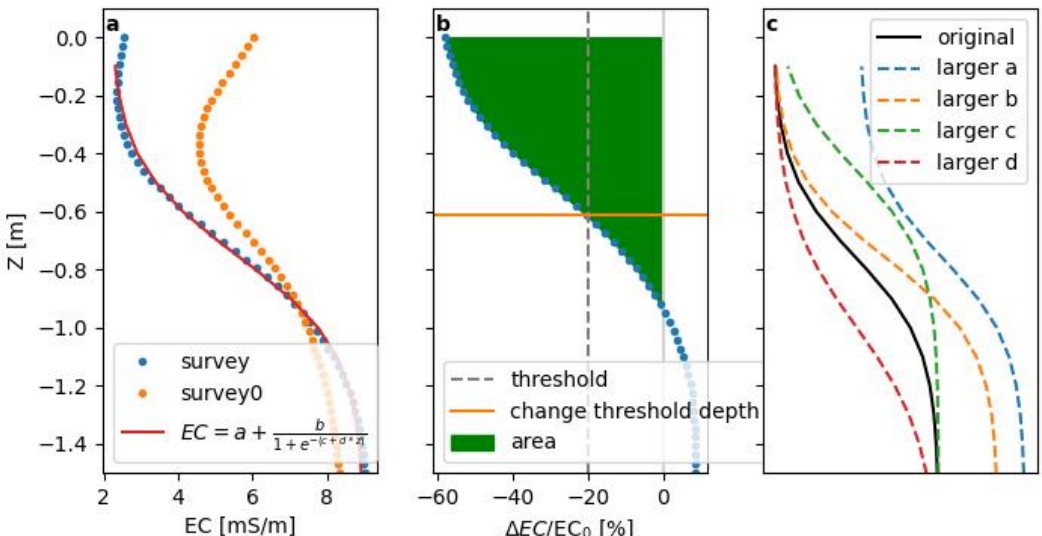

**Figure 8.** Example of indicators computed either on (a) electrical conductivity (EC) profile or (b) its relative difference ($\Delta EC/EC_0$) from the reference survey (survey0). Subplot (c) shows how the sigmoid function change with respect to an increase of each of its parameters.





To enable users to interact with the dataset, explore and compare profiles in space, depth and time, all processed data from
the ERT pipeline was summarized into an HTML report with interactive Bokeh (https://docs.bokeh.org, v3.4.0, last accessed
2024-03-25) figures (available in the Gitlab repository).

## 3 Results

### 3.1 Apparent resistivities

The first question of the POC experiment was whether the electrical resistivity measurements are sensitive enough to detect
subtle differences in water depletion patterns and strategies between contrasting genotypes of the same crop species. Figure 9
shows the pseudo-section of the ERT transect B in the control field. Raw apparent resistivities reflect consistent patterns related
to different root water uptake patterns of the three investigated genotypes in different plots along the transect. For instance,
Hermes and Pro_1 take up more water already earlier in the growing season than Lenka. We observe similar trends in the part of
the transect with plant rows longitudinal to the ERT transect as in the part with rows crossing the transect.

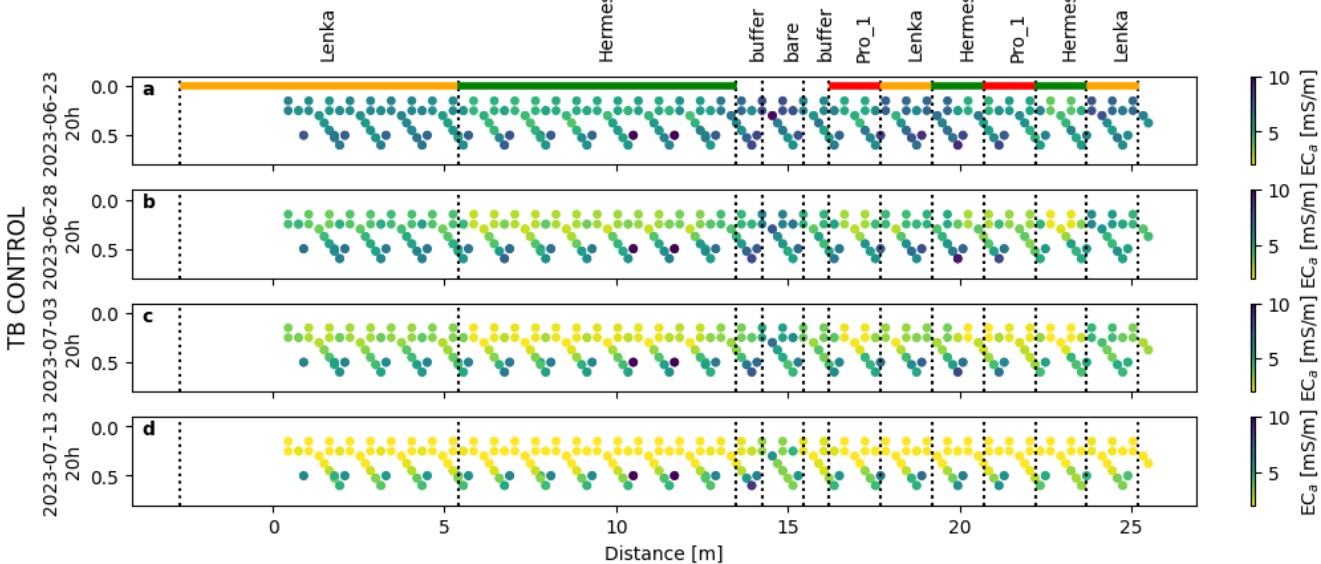

**Figure 9.** Evolution of pseudo-section for transect B in the CONTROL field. Note that the depth on the vertical axis is not the actual depth of
the reading, but rather an estimation of the depth above which most of the signal originate (a pseudo-depth) based on the electrode location
and measurement geometry.

The impact of the drought treatment is clearly visible in the apparent conductivity values (see ECa distributions Figure 10).
This confirms the capability of ERT to monitor the dynamic impact of both treatment and genotype on bulk soil electrical
properties. This means that even without further inversion or other data processing, a statistical test can be executed to





discriminate between genotypes in terms of the impact of drought on belowground behaviour. Figure 10 shows the distribution
of the apparent conductivity values of each treatment x genotype combination. We performed an ANOVA (analysis of variance)

255 with the *statsmodels* Python package v0.14.0 (Seabold and Perktold, 2010) considering two factors: genotype (Hermes, Lenka,
Pro-1) and treatment (drought, control). The effect of drought is significant (p-value <0.05) for shallow pseudo-depths at
all 4 timepoints and for the deepest pseudo-depth only in mid-July. The genotypes show significant differences for shallow
pseudo-depth, but not for the deeper depths. A root sampling campaign, just after harvest, has shown that there are little roots
present at depths below -0.5 m in any of treatment x genotype combinations, with Pro-1 showing the most roots in the deepest

260 layer, especially under drought (data not shown). This might explain why did not find a significant difference in ECa below that
depth.

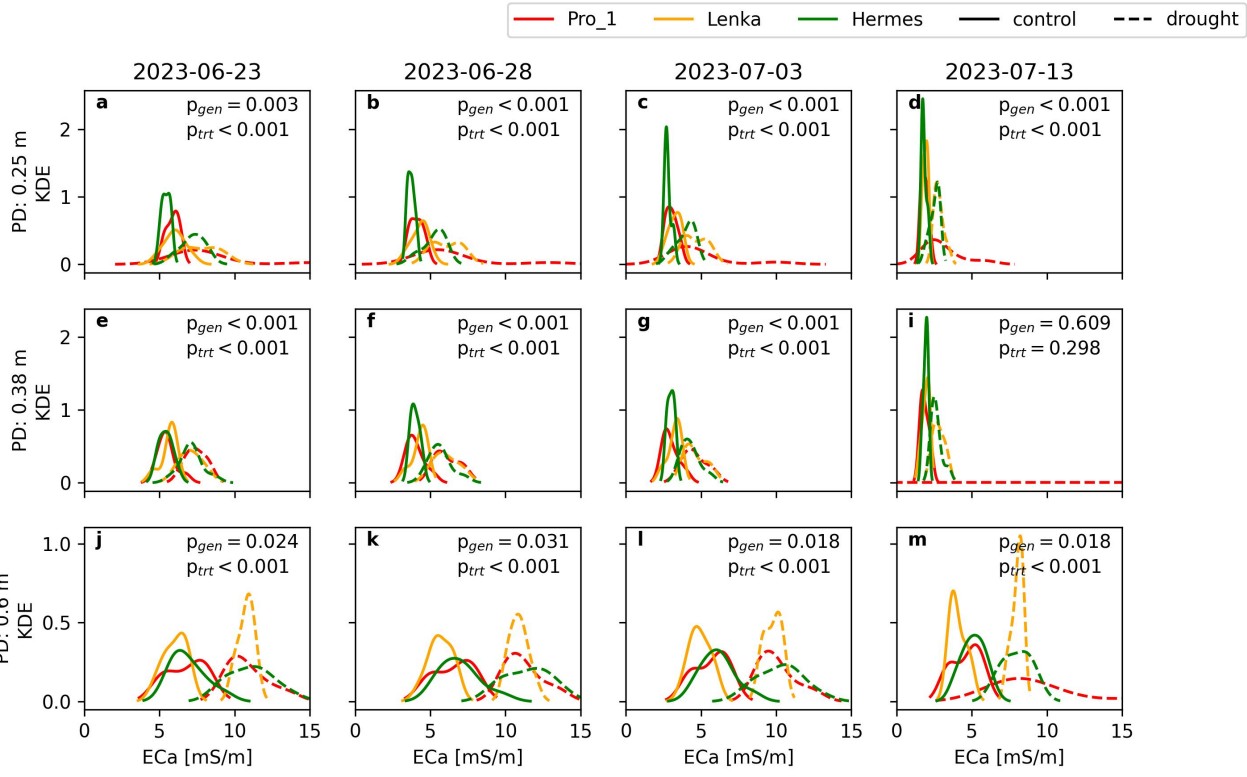

**Figure 10.** Kernel density function of all apparent conductivities for three different pseudo-depths (PD) along with ANOVA p-values for the
genotype factor ($p_{gen}$) en for the drought/control factor ($p_{trt}$). All p-values are rounded to three decimal places. The interaction term was not
significant (>0.05) for all cases. The subplots at different depths do not share the same vertical axis.

Apparent conductivity data do not disclose information about specific depths of water depletion (but rather pseudo-depths).
This makes it difficult to relate the data to plant traits such as rooting depth, root density, or depths of water uptake. To obtain
depth-explicit information, a data inversion or other advanced data processing such as coupled modelling and/or machine





learning is necessary. In addition, it is not possible to use the pedophysical relationships from the control pit to convert apparent EC to depth-specific soil water content. Indeed, the apparent resistivities are depth-weighted integrative measurement while soil water content from sensors is depth-specific. Only inverting the apparent resistivities will enable us to obtain depth-specific EC values which can then be converted to soil moisture with a pedo-physical relationship.

## 3.2 Inverted transects

Figure 11 shows the inverted transect B (CONTROL) at three moments in June during a period with little rain. Differences in soil water depletion between genotypes are apparent. For instance, Lenka took up less water than Pro-1 and Hermes. The figure also illustrates how the drying front tends to increase with time. Water depletion is mainly caused by the crop, as the bare plot in the middle does not show the deep drying pattern.

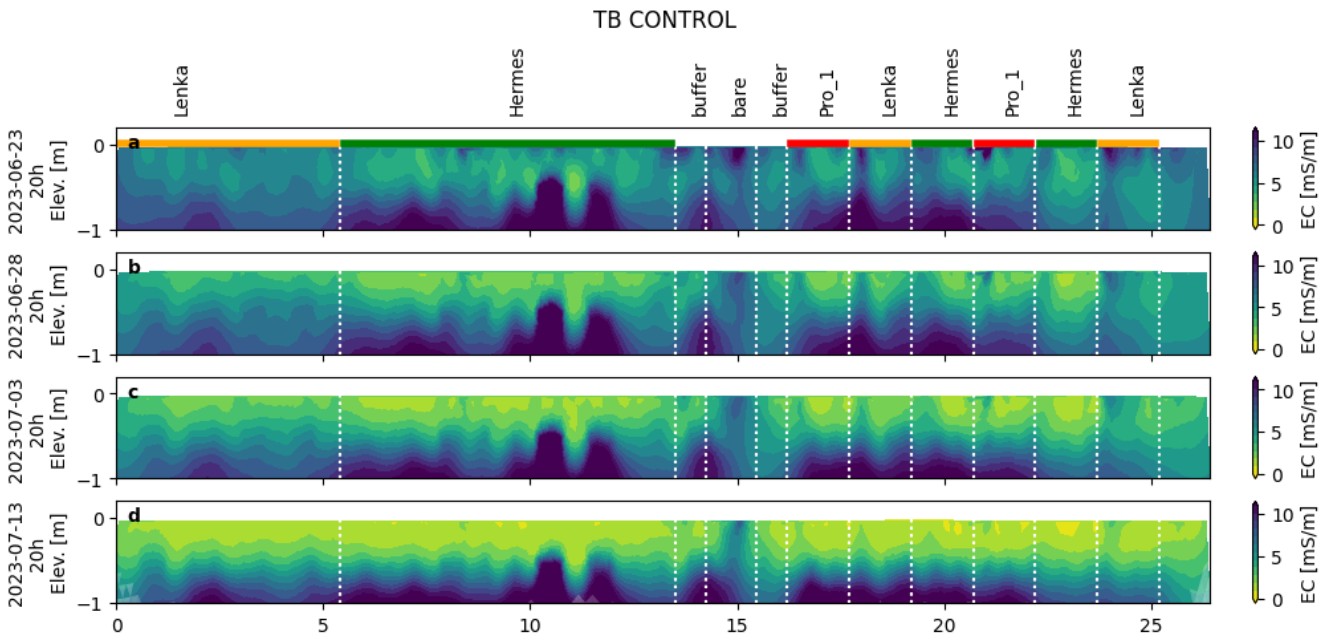

**Figure 11.** Evolution of the selected inverted transect B in the CONTROL field. Plot positions are denoted with vertical dashed line and horizontal coloured lines.

## 3.3 Profile evolution

From the inverted section, we extracted average profiles for each plot and then merged per genotype by taking the average profile of all plots. By subtracting the average background profile from this, we can depict the %-change occurring over the profile as compared to the background. Drying under the CONTROL is faster during the early natural drought, but more limited in time than for the DROUGHT treatment in which an artificial drought was generated using the rainout shelter. After the rainfall events





of July, the CONTROL field becomes wetter than at the start (blue band starting in August, Figure 12) and only superficial
280    drying was observed after. Zooming into the difference between genotypes in the same treatment, we can see that the amount of
water of the rainfall in the beginning of august infiltrated less (deep) under Pro-1 than under Lenka, which could be due to a
different transpiration rate or depth of water uptake between the two genotypes.

**Figure 12.** Evolution of profiles of inverted, temperature corrected electrical conductivity (EC) for the three genotypes under both treatment
(CONTROL and DROUGHT) during the growth season. The rainfall deficit is shown in subplot a for both CONTROL (plain black line) and
DROUGHT field (dashed black line). The period of the drought application is indicated by the yellow shaded area. Missing data are indicated
by white bars at that date. The white line shows a -50% change while the dark blue line shows a 0% change.



## 3.4 Indicators

Figure 13c-f shows the evolution of four selected indicators computed from the inverted electrical conductivity profile and
the profile of change in conductivity with respect to background for different genotypes. We present the indicators alongside
meteorological conditions (Figure 13a) and crop development as represented by the UAV based phenotyping SAVI vegetation
index (Figure 13b). The indicators differ clearly between the CONTROL and the DROUGHT treatment. From August onwards,
genotypes submitted to the DROUGHT treatment exhibit larger drying patterns than in the CONTROL. This is apparent
from the Drying Area (DA) indicator, which resides much longer in a zone of big water depletion (large negative value) in
DROUGHT than in CONTROL, and this for all genotypes. This is because in the CONTROL, rainfall replenishes the soil
moisture profile. The effect of crop water uptake is therefore partially undone. The DA even becomes positive at some moments
in time, which means that the soil was wetter then than at the start of the experiment. The difference between treatments
decreases from September onwards. Since the shelters were removed, both treatments were receiving rain again and the crop
reached physiological maturity. Where DA is a robust indicator which is smooth over time because it represents an integration of
the whole profile, the other indicators are more noisy. The Drying Depth (DD), for example, represents one specific point in the
profile intersecting with the 20% threshold, which is much more sensitive to slight changes in the profile. We performed ANOVA
tests on all timesteps of all four indicators and indicated at which time during the year the genotype effect and/or the treatment
effect is significant using grey dots at the top and bottom edge of the subplots. Significant difference between genotypes (dark
grey) are mostly observed in the beginning of the growing season, which correlates with the pseudo-sections shown in Figure 9.
Just like with vegetation indices, curves could be fitted to the time evolution of these belowground indicators to assess rates of
drying or similar properties.





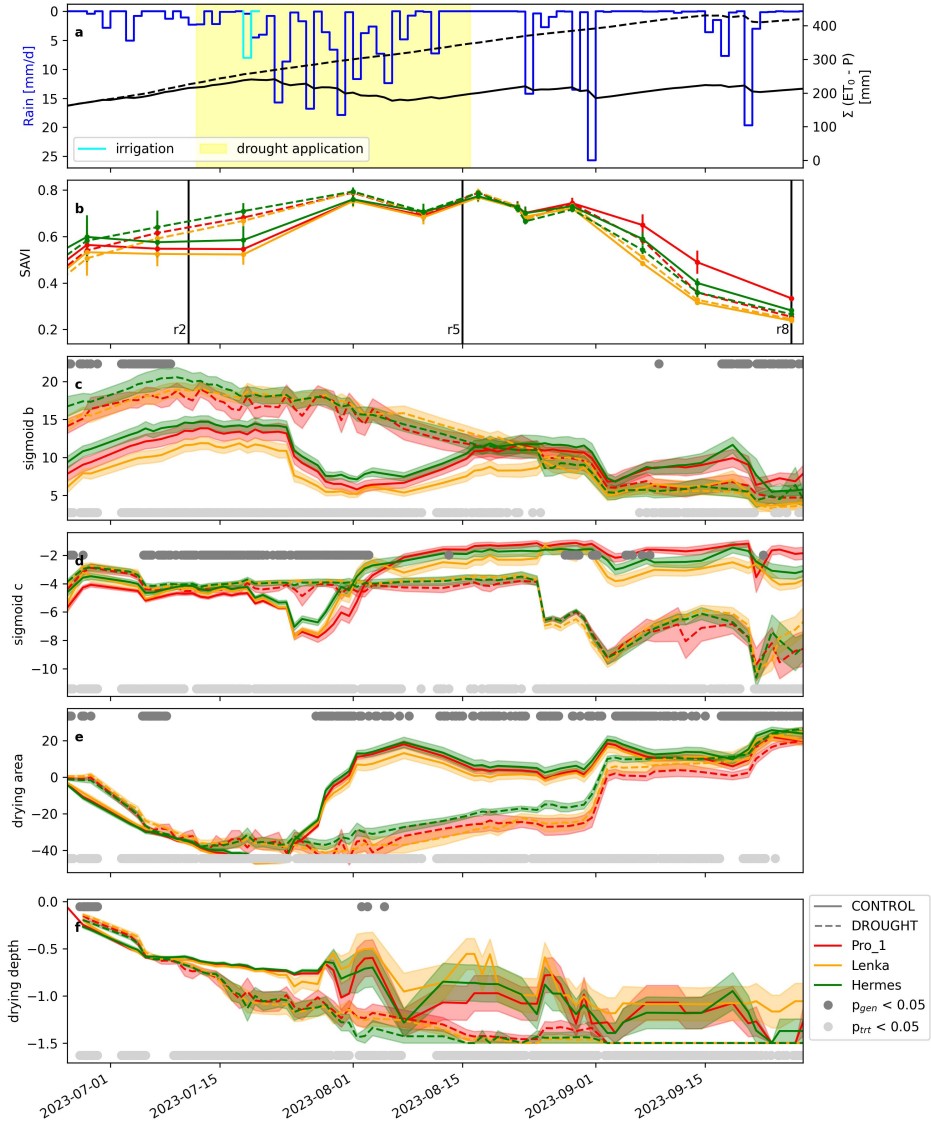

**Figure 13.** Example indicators and their evolution for the three genotypes as a function of the treatment applied (plain line: CONTROL, dashed line: DROUGHT). a) Rainfall and precipitation deficit, b) SAVI = Soil-adjusted vegetation index and important growth stages (reproductive stages r2, r5 and r8), c-d) parameters b and c of the sigmoid fitted on the electrical conductivity profile, d) Drying Area indicator f) Drying Depth indicator. For the subplots with indicators (c-f), an ANOVA was performed. The grey and light grey dots show when the p-value for the genotype factor and the DROUGHT/CONTROL factor was below 0.05, respectively.

In Figure 14, we zoom in on the genotype differences in our 4 selected indicators at 2 specific moments in the crop growth: flowering (r2), just before the DROUGHT application, and seed filling (r5), just after the DROUGHT application. CONTROL is shown in blue and DROUGHT in orange. In sigmoid c we see the clearest genotype effect at July 11th. This parameter reflects the inflection point of the sigmoid, therefore related to the drying depth.





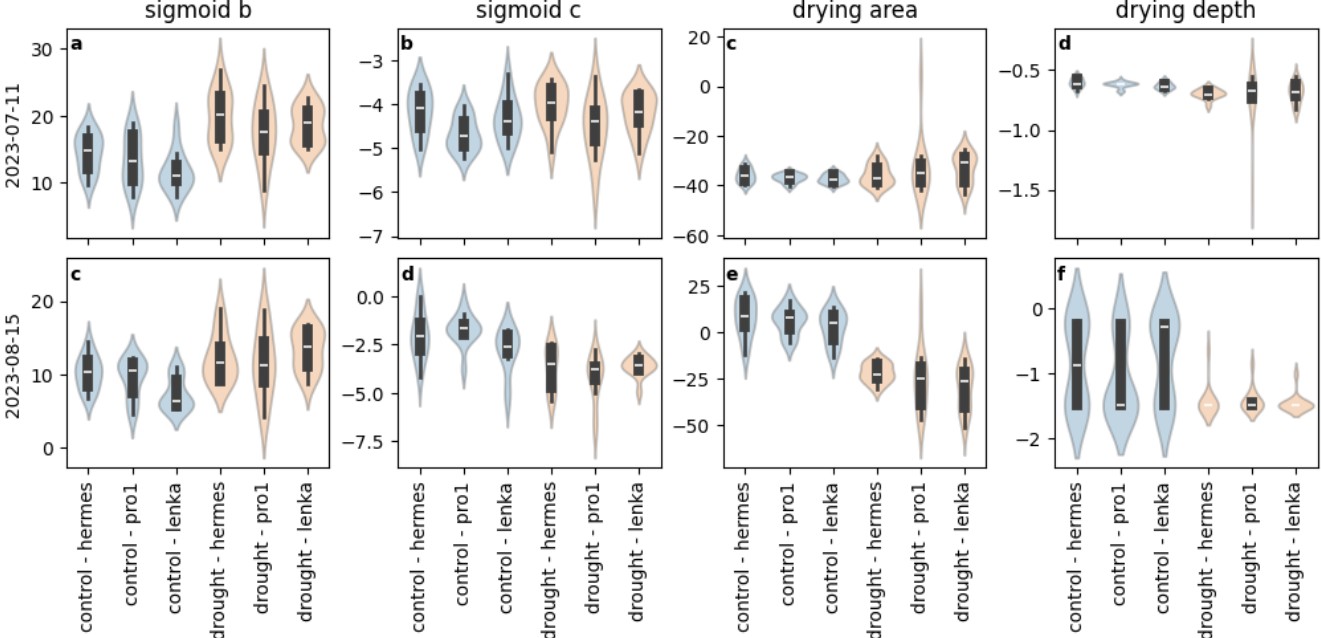

**Figure 14.** Violin plots showing the distribution of the indicators calculated from the electrical conductivity profiles for the 3 genotypes in CONTROL and DROUGHT.

# 4 Discussion

The proof-of-concept experiment with the HYDRAS infrastructure in 2023 established a dataset at high spatial and temporal resolution. This dataset characterized differences in belowground behaviour between control/drought treatments and soybean genotypes, under realistic field conditions in a real, living soil. We have shown that ERT is not only capable of detecting

differences between crops or treatments as previously done in literature, but also has sufficient precision to distinguish between genotypes of the same crop. We highlighted the potential to automatically derive phenotyping indicators related to dynamic belowground plant traits. To our knowledge, the only permanent infrastructure which is capable of monitoring water depletion and root system activity at the same spatio-temporal resolution is the sEIT installation at the Selhausen minirhizotron facility in Selhausen, Germany (Weigand et al., 2022). The sEIT system there consisted of 40 electrodes (0.25 m electrode spacing)

installed across three agricultural test plots (each 3.75 m width). This site was not designed for high-throughput field phenotyping. It does not have rainout shelters, but did allow testing and validating many of the available techniques over several years of operation, yielding an impressive open subsoil data set (Lärm et al., 2023). In that installation, not only electrical resistivity, but also chargeability properties are measured, which is a meaningful addition to the information which can be derived from resistivity. Chargeability was shown to be more directly linked to root biomass, which opens up new possibilities (Weigand and

Kemna, 2017). However, accurately measuring these properties with sufficient data remains challenging.



Weigand et al. (2022) highlighted the need to further investigate innovative ways to assess uncertainties without compromising measurement time and more importantly, ways to incorporate that information in the data processing and interpretation. This is also the case for the current data set, which exhibits error levels which change over time. We do take them into account in the inversion, letting the error model vary over time, but we did not yet assess how this affects the precision of our derived

indicators. Nevertheless, the automated, online monitoring also directly allows the user to spot and resolve inevitable field issues instantaneously. The SSI system provides email alerts for the operators using basic and more advanced warning thresholds based on system status (battery, temperature, connectivity) and measurement metrics (contact resistance, plausible values, . . . ). This makes it possible to minimize error sources and data gaps.

The HYDRAS ERT setup has the advantage of having both a surface and a buried cable to enhance the sensitivity and

resolution in the root zone. Based on this configuration, quadrupole along the surface line, along the buried line and between the surface and buried line were added. However, the combination of cm-inaccuracies in the placement of the belowground cable with small electrode spacings resulted in inversion difficulties. Quadrupoles with injection electrodes in the surface line and potential electrode in the buried line (or inversely) led to inversion artefacts close to the electrodes when inverted. We tested the approach of Wilkinson et al. (2015) to include electrode location as part of the inversion, but this resulted in too many degrees of

freedom in our case. However, simulating the data with slight electrode displacements (longitudinal or lateral) resulted in similar artefacts, which confirms the hypothesis. Further investigation is necessary to fully exploit the potential of the subsurface cable without compromising the inversion results.

The HYDRAS POC2023 dataset is not only one of the rare freely available datasets resulting in defined belowground phenotyping indicators which can be related to crop traits, but also the first to develop the belowground phenotyping data

acquisition and processing pipeline in a fully automated way at field scale. This resulted in a standardized, reproducible and high-quality data set and associated processing scripts. Although several attempts exist to make geophysical data sets more FAIR (Findable, Accessible, Interoperable and Reproducible), the agrogeophysical community is far from reaching a community data model or reporting standard. Initiatives have been launched, such as the CAGS, but often become inactive or phase out after project funding stops. Others, such as the REDA package, remain, but are not widely adopted. In addition, these existing

initiatives do not yet seek much compatibility with data standards from the research communities which seek to use the data, such as the phenotyping or precision agriculture communities. The POC2023 and following open data sets from HYDRAS seek to bridge that gap and open up the discussion on the specific needs of users in distinct use cases. For the HYDRAS pipeline, we explored several data models from different scientific communities and came up with a data model proposal largely compatible with the eLTER and the MIAPPE standards.

The infrastructure is not only a crucial asset for the phenotyping and breeding community. 2-D and 3-D, high resolution automated monitoring of water flow, solute and heat transport processes in the undisturbed soil-plant-atmosphere continuum under agricultural land use provides several exciting opportunities, especially since the mobile shelters provide crucial control on the top boundary condition of the soil. One of the opportunities is improving the understanding of how soil health and its management drives plant performance in agroecosystems. As Carminati and Javaux (2020) and Abdalla et al. (2021) have

highlighted, soil, and more particularly the rhizosphere, is a major driver of the plants response to drought. However, studying



this in field conditions remains challenging. The HYDRAS infrastructure, complemented with ground-truthing data of small scale processes in the rhizosphere or at the level of plant organs, could help better understand how agricultural practices can optimize the soil-plant interactions. Without aspiring to be exhaustive, potential fields of research tackled in this infrastructure could be the investigation of (preferential) flow and transport of agro-chemicals, the impact of agricultural management practices

on soil health in its various dimensions and on water use efficiency of crops, the effects of irrigation with different types of water sources on soil salinization, etc. In addition, this infrastructure enables to further explore the drivers and multiscale nature of the pedophysical relationship and the uncertainties associated with it. Taking into account the heterogeneities in the pedophysical relationships can improve our ability to detect differences between genotypes. (e.g. Blanchy et al. (2020b)). A full analysis of the uncertainties associated with the pedophysics is out of the scope of this paper but will be actively investigated in future work.

In short, this infrastructure provides new and exciting opportunities for the broad soil science community.

## 5 Conclusion

The HYDRAS POC2023 experiment has shown that ERT monitoring at high spatio-temporal resolution offers unprecedented capabilities for reproducible and accurate belowground field phenotyping. The raw apparent conductivity data clearly show differences between treatments (drought/control) and soybean genotypes, highlighting the information content in the data.

Inversion allows to further fine-tune the information to depth-specific data, which is then used to develop phenotyping indicators related to specific plant traits of interest. Further improvement is desirable to quantify the uncertainties added in each step and how these propagate through the entire acquisition and processing pipeline to the final indicators, since this influences the interpretation of the data and the power of statistical tests. Various approaches are present in literature, including Bayesian inversion techniques, joint inversion and coupled inversion using soil-plant models, which can now be tested on the big

phenotyping dataset.

*Code and data availability.* The notebooks and data used to generate the figures in this paper can be found at https://gitlab.ilvo.be/hydras/ert-paper (TODO ADD DOI upon publication). The dataset of POC2023 can be found here (ADD DOI from ZENODO upon publication

## Appendix A:  Field management activities



**Table A1.** Field management activities

| Date | Action |
| --- | --- |
| 2023-05-19 | Seedbed preparation |
| 2023-05-23 | Sowing soybean (row spacing 25 cm, plant spacing 6.2 cm, sowing depth 3.5 cm) |
| 2023-05-24 | **Sowing soybean on ERT fields** (parallel and perpendicular sowing direction) |
| 2023-05-25 | Soil herbicide application |
| 2023-05-26 | Installation ERT cables and electrodes |
| 2023-05-26 | Netting |
| 2023-05-30 | First germination observed |
| 2023-06-12 | Removing net |
| 2023-06-13 | Weeding |
| 2023-06-21 | Spraying |
| 2023-06-21 | **Start ERT monitoring** |
| 2023-07-12 | **DROUGHT START rainout shelters placed above 4 plots (a1,b1,c1,d1)** |
| 2023-07-19 | Control plots irrigated (8 $mm.m^{-2}$) |
| 2023-08-16 | **DROUGHT STOP rainout shelters removed from 4 plots (a1,b1,c1,d1)** |
| 2023-08-29 | Pesticide application: Acaricide floramite was sprayed on all fields (dose: 0.4 $l.ha^{-1}$) |
| 2023-10-04 | **Harvest** |

## Appendix B: Measurement sequence

The sequence of quadrupole used is composed of a "plot sequence" that is moved along the surface and buried line of a transect. The "plot sequence" includes 12 surface electrodes spaced 0.3 m (electrodes 1 to 12) and 6 buried electrodes spaced 0.6 m (electrodes 13 to 18). The "plot sequence" contains one injection in the surface electrodes with potential readings between the injection dipole (gradient), outside on the surface and the buried lines (dipole-dipole). Another injection is done with a buried pair of electrodes and similar potentials are collected. The "plot sequence" is repeated every 4th surface electrode (2 buried

electrodes) along the transects to form the final sequence used. This measurement sequence was chosen as compromise between spatial and temporal resolution. The sequence includes all reciprocal quadrupoles.





**Table B1.** Plot sequence repeated along each transect. A and B are injection electrodes, M and N are potential electrodes used to measure voltage. Electrodes 1 to 12 are surface electrodes with 0.3 m spacing. Electrodes 13 to 16 are buried electrodes with 0.6 m spacing. Electrode 1 is on top of electrode 13.

| A | B | M | N |
|---|---|---|---|
| 1 | 6 | 2 | 3 |
| 1 | 6 | 3 | 4 |
| 1 | 6 | 4 | 5 |
| 1 | 6 | 7 | 8 |
| 1 | 6 | 8 | 9 |
| 1 | 6 | 9 | 10 |
| 1 | 6 | 10 | 11 |
| 1 | 6 | 11 | 12 |
| 1 | 6 | 13 | 14 |
| 1 | 6 | 14 | 15 |
| 1 | 6 | 15 | 16 |
| 13 | 16 | 2 | 3 |
| 13 | 16 | 3 | 4 |
| 13 | 16 | 4 | 5 |
| 13 | 16 | 6 | 7 |
| 13 | 16 | 14 | 15 |
| 13 | 16 | 17 | 18 |
| 2 | 3 | 1 | 6 |
| 3 | 4 | 1 | 6 |
| 4 | 5 | 1 | 6 |
| 7 | 8 | 1 | 6 |
| 8 | 9 | 1 | 6 |
| 9 | 10 | 1 | 6 |
| 10 | 11 | 1 | 6 |
| 11 | 12 | 1 | 6 |
| 13 | 14 | 1 | 6 |
| 14 | 15 | 1 | 6 |
| 15 | 16 | 1 | 6 |
| 2 | 3 | 13 | 16 |
| 3 | 4 | 13 | 16 |
| 4 | 5 | 13 | 16 |
| 6 | 7 | 13 | 16 |
| 14 | 15 | 13 | 16 |
| 17 | 18 | 13 | 16 |



## Appendix C: Overview of inversion quality indicators

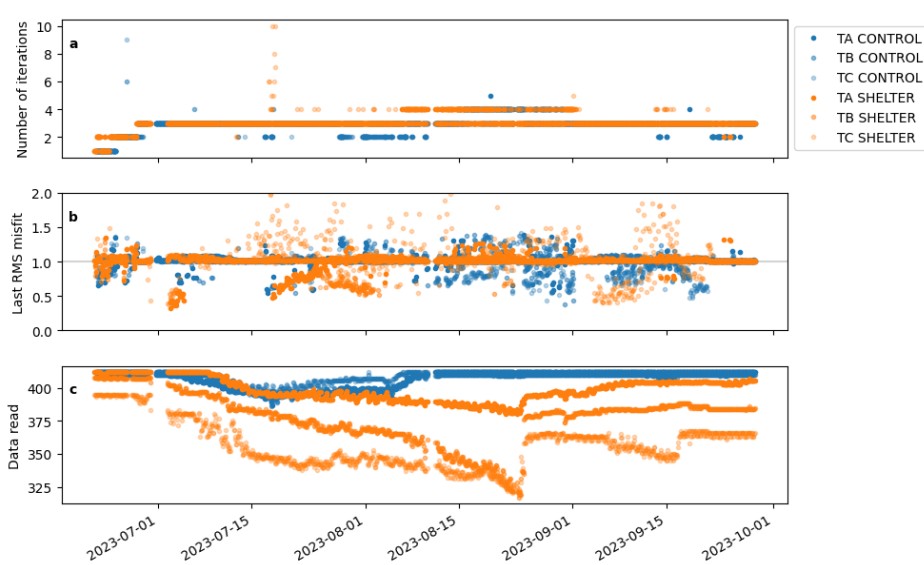

**Figure C1.** Overview of inversion quality indicators (a) number of iterations, (b) final root mean square (RMS) misfit and (c) number of data read (i.e. number of data after filtering that are actually inverted).

*Author contributions.* Conceptualization: GB, SG, WD, TDS,

Formal analysis: GB, SG

Data Curation: GB, SG, WD

Writing - original draft: SG, GB

Writing - review & editing: all authors

Project administration: SG, IRR, PL

*Competing interests.* The authors declare no competing interest.

*Acknowledgements.* The HYDRAS open access infrastructure is a joint effort. We would like to thank colleague Filip De Brouwer and the entire O&I field team of ILVO for the planning and execution of all necessary field operations as well as the above ground High-Throughput Field Phenotyping using UAVs equiped with different sensors. We also thank FWO for funding this research infrastructure in their 'International Research Infrastructure (IRI) call' with the project Emphasis-Belgium: phenotyping the future crops (I002121N) and the





Flemish government for their additional support under the project FutureAdapt (VV028). Guillaume Blanchy is a Research Fellow of the
Fonds de la Recherche Scientifique – FNRS (CR: 1.B.044.22F).



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
