# Peer review of "Closing the phenotyping gap with non-invasive belowground field phenotyping"

_EGUsphere, 2024_

## Author Response (AR1)

**RC1**

| Comments | Answers/Proposed changes |
|---|---|
| L57 Consider adding a couple of references for readers that are less familiar with this.

For example, *Extreme sensitivity of crosshole electrical resistivity tomography measurements to geometric errors* by Wilkinson et al, 2008; and *Mitigation of installation-related effects for small-scale borehole-to-surface ERT* by Ochs et al, 2022. | We added the two suggested references as they illustrate well errors that can arise with borehole electrodes. |
| L66 Consider adding *An overview of multimethod imaging approaches in environmental geophysics* by Wagner and Uhlemann, as a general reference here. | We agree. This gives a nice overview of the options. |
| L89 "rain-out shelters" is then spelled rainout shelters. I think both may be right, just for consistency. | Agreed. We homogenized all to rainout. |
| L97 "en" should be "and"? | Corrected. |
| L106 how these petrophysical relationships were derived, should be clarified later | We added 'by curvefitting to the soil moisture – electrical conductivity data from the pit' already here. |
| L112 "Dryer soil, where roots are extracting water, has a larger resistivity (lower electrical conductivity) than wetter soil around the root zone." I agree, I would add that this is because the root-induced water dynamics dominates other effects, e.g., static and direct contribution of the root biomass vs RWU. You mentioned above these effects, I think connecting them here would clarify this assumption to the general audience. | We added: 'The effect of root-induced soil moisture changes is generally much larger than other, direct effects of the root biomass.' |
| L123 Missing reference / link to figure 1 | Corrected |
| L140 The ERT setup is nice, well designed, and robust, also thanks to the many electrodes and reciprocal measurements. I find the one-channel system to be the only limiting aspect here, for the obvious trade-off between time and number of measurements (and thus spatial resolution, temporal resolution, and data quality stability). However, pushing new and cheaper instruments for long-term monitoring is also central | We added 'There is an obvious trade-off between acquisition (slower for single channel than for multi-channel device) and instrument price (cheaper for single channel than for multi-channel devices). The cheaper instrument price of the SSI single channel device allows for multiple devices in the field and simultaneous data acquisition of multiple fields.' |

| | |
|---|---|
| here, and thus in line with the paper goal. Consider discussing these points. | |
| L142-145 I agree on the need to verify the impact of the plant-ert alignment, nice side investigation. The supposed causes are reasonable in my opinion; with systematically you also mean independently from their geographical orientation (i.e., N-S vs E-W)?

Was there some extra space - gap (bare soil or grass) between the different varieties, or only the standard row spacing? | The yield was lower in the smaller plots of the transversal (E-W) orientation.
Between the different varieties, there was no gap (see fig. 2a). The transition is continuous. |
| Fig3 b and c, Control and pit are very different; what are the implications for the underlying relationships etc.? | Control and pit are mainly different because the pit is kept bare (with a bit of weeds) and control has a crop and therefore dries out more. The soil moisture status also affects the heat transfer properties and therefore the soil temperature (especially at the surface). We did a spatial assessment of soil properties and the pit is a good representation of the type of soil present in the field, although of course spatial heterogeneity is present and having several pits and/or complementary pedophysical data on soil cores from several places would be even more accurate. Overall, the pit data is a good estimate of the overall behavior of the different horizons. Due to the lack of a crop, we have less data in the dry range of the relationship than what occurred in the field. There the relationship is less robust, but we continue to measure over the years and hope to thus establish a good knowledge of the entire range.

We added the following in this section:
'As the soil pit was bare, and the fields were sown with soybean, the evapotranspiration was higher on the fields than in the pit, which resulted in lower soil moisture contents in the field than in the pit. The pit should therefore not be used to assess the field status of soil variables, but rather as a field-scale source of data on soil hydraulic, thermal and pedophysical functions. ' |
| Fig3b, the drought VWC seems to respond more slowly on surface than at depth in summer and fall, check if this is ok. | Only the soil pit has sensors at different depths. Black is most superficial and reacts the fastest.

The orange and blue hues just denote different individual sensors all installed at the same depth (namely vertically from the top with the center of the measurement volume at 0,15m depth).

All is therefore ok with the figure, we think. |

| | |
|---|---|
| | The caption says:
'The yellow band indicates the length of the applied drought treatment with the rainout shelter. (b) Soil moisture in the calibration pit at four depths (grey hues) and in the drought (orange hues) and control fields (blue hues) at 0.15 m depth (different hues = different locations in the field).'

Is this now sufficiently clear? |
| L188 – L193 Yes, I agree that the quadrupole geometry (and associated geometric factor) is a key aspect here. Ideal – perfect placing is impossible at this spacing, and local changes (e.g., small electrode position and contact) over the monitoring period may have surprising effects too, on weak quadrupole geometries. As commented above, there are some references on this, being your geometry equivalent to the XH and IH conditions in borehole measurements, right? I like the numerical testing, adding some references would provide a bit of background. | Indeed, we added 'More background on the effect of electrode misplacement on ERT accuracy can be found in amongst others Wilkinson et al. (2010); Uhlemann et al. (2018); Oldenborger et al. (2005).' |
| L196 "The reciprocal errors were not used for filtering, but as a weight during the inversion. We fitted a power-law error model for each survey on a transect using the binned reciprocal errors (Koestel et al., 2008)." Wouldn't this keep clear outliers, with an error based on the general dataset fitting?

In my opinion, it makes sense to directly remove clear outliers as soon as possible (say rec. err > 50 %?), because the general fitted error would not be representative for these outliers, but also to avoid affecting the fitting procedure. I think this was also done in the cited work too.

I see that you considered stacking, geometry, neg. apparent resistivity, and contact resistances; these would likely capture most of the clear outliers, limiting the effect of the above choice. Overall, I think the quality assessment is quite extensive and innovative considering the scale of the monitoring. | We generally agree with the reviewer. For this study, the filtering we applied already removed most of the outliers and reciprocal errors were re-computed afterwards. Such that if a reciprocal error was large because the normal or reciprocal quadrupole was an outlier, this reciprocal error will not be considered in the fit.

We wanted to avoid filtering directly on the reciprocal because this will remove both normal and reciprocal quadrupole (as we don't know which one is bad – both can be bad too). By first filtering on all quality indicators (contact resistance, stacking error, …) related to the acquisition, we can retain more quadrupoles.

In addition, the use of a log-bins during the power-law fit, reduces the effect of large reciprocal error outlier on the power-law fit (which wouldn't be the case with an envelope fit or a fit on all data). However, we agree that filtering on the reciprocal error can be needed.

We modified the text as follows: 'In this study, the reciprocal errors were not used for filtering. Most outliers were already removed by the previous filters. However, filtering on reciprocal error might be needed in more noisy surveys ". |

| | |
|---|---|
| L232 italic font for the parameters *a* and *b*, as done for the successive *c* and *d*. | Corrected |
| Fig8 Consider adding the x label to subplot c; it is just the same as b, right? | Added. Note that we didn't specify the ticks as this subplot c is only illustrative and shouldn't be interpreted quantitatively on the axis.

Note Fig7 has also been updated. There was an error in the legend. |
| Fig9 I agree that time differences can nicely be observed already in the pseudo-sections, also very good from a data management point of view. Having some extra points would help (as discussed above on the ERT acquisition). Are these all the quadrupoles or just some selected ones? | These are all quadrupoles indeed, so no points can be added. To save time, we had to cut down on overlapping quadrupoles and only keep the essential ones. |
| L265 This closing remark is needed. Maybe mentioning some aspects already at the beginning of the section would further highlight the contrast between "whether the electrical resistivity measurements are sensitive enough to detect subtle differences in water depletion patterns and strategies between contrasting genotypes of the same crop species." and the successive goals that rely on the ERT inversion. Guiding a bit more the general reader.

Consider also giving a couple of simple/practical examples on what it means "apparent resistivities are depth-weighted integrative measurement". For example, strong evaporation on the surface would lead to larger apparent resistivities in the deeper parts too, which could be confused for RWU; or rain/irrigation would affect the deeper parts too. | We've added some text in the beginning of the paragraph to better guide the reader from the pseudo-section to the statistical distributions:

"To further detect the subtle difference between genotypes, we take advantage of the statistical design of the study and look at the ECa distributions.".

We have added the example: "For example, large drying/wetting at the surface will decrease/increase the apparent conductivity in the surface but also, to a lesser extent, for quadrupoles with deeper measurement volumes." |
| L281 capitalize "august" | Corrected |
| Fig12 and L281 I don't see the difference at the beginning of August, do you mean September in the DROUGHT treatment? The CONTROL time series look very similar to me, also relative to the pseudo section. Consider better highlighting the differences. | Indeed, it is very difficult to see a difference from the figure or the pseudo-section. We decided to remove this sentence and use other figures (like the indicators) to highlight the difference between genotypes. We replaced it by "From the colour gradient, difference between genotypes remains difficult to see, hence why indicators were derived." |
| L287 "August onwards" more end of July? Considering the rain event between July 15 and August 1. For | Yes correct. Corrected. |

| | |
|---|---|
| example, the drying area increases significantly during this period but stabilizes in August, correct? | |
| L296 consider referencing Table1 when discussing DA and DD. | Reference added. |
| Fig14 Were the sigmoids fitted to the actual conductivity profiles or to their ratio with respect to the background (i.e., EC or deltaEC)? Considering figures 11 and 12 it seems that the sigmoid may be more suitable to describe the conductivity itself. For example, is the described sigmoid suitable for the drier-wetter-drier profiles of deltaEC shown by the CONTROL after August in figure 12? Please clarify this point. That said, fitting the actual VWC changes or the distribution of the water potential will be the necessary successive step, and the sigmoid should be suitable for this. Currently, both EC and deltaEC profiles would fail as proxies (and possibly deviate from a sigmoid) in the presence of significant texture stratification or anyway variable pedophysical relationships, right? | The actual conductivity profiles as specified in fig8a legend. While the original idea was to fit sigmoid to the DeltaEC, indeed the drier-wetter-drier pattern observed in the CONTROL didn't make the fit possible as you noted. We agree that efforts should be made to improve this fit and also apply it to the DeltaEC. Soil stratification remains a challenge, as observed change seems also to be affected by this stratification. Maybe more advanced functions like piecewise sigmoid or spline could help here. Ideally, this fit is also done on the estimated WC profiles, so that the effect of other variables/properties is dealt with in the pedophysics.

The fit presented in this work is, in this regard, a first attempt at deriving meaningful indicators to show the potential and will certainty be improved in the future. |
| L324 "but we did not yet assess how this affects the precision of our derived indicators." I agree, this could be interesting. | We added it would be nice for future research. |

**RC2**

| Comments | Answer/Proposed changes |
|---|---|
| Line 30. Up to here, the text was very well written. However, I think you can do better on the introduction of the ERT method. I was missing some keywords here, such as geophysical, imaging, and perhaps inversion. Perhaps you can improve. | We agree that a bit more explanation might be useful for researchers unfamiliar with ERT. We added some text to deal with this comment and the next.

'' |
| Line 33. A key challenge remains that ERT is not only sensitive to soil moisture. I would suggest to emphasize this early on. | Addressed together with previous comment. See new text above. |
| Line 35. The sentence starting with Wasson et al. distracted me from the line of argument. | Done. |

| | |
|---|---|
| Consider rewriting by just citing Wasson et al. for the following statement. | |
| Line 72. I would also mention the limited descriptions of the used inversion approaches, and the determination of the regularization strength. I always thought that there a range of subjective decisions during inversion that are not sufficiently communicated. | Agreed. We added a bit more on this in the paragraph on inversions above:

 'Developing robust inversion algorithms with clear descriptions of used approaches and applied regularization strengths remains importnat. Improving the transparancy and accuracy of the inversion process is necessary to develop standardized processing pipelines and resulting indicators for breeders. ' |
| Line 73. For the mentioned initiatives, it would be good to add references to a report, website or publication. | These are hard-coded as URLs in the latex document. We will see with the editors whether this is ok or whether the url should appear as text in the manuscript. |
| Line 108. Should this be near the soil surface? If not, consider rewriting... | Done. |
| Line 112-115. This description of how ERT provides information on root water uptake can be improved. | We changed the text as follows:

 'In HYDRAS, Electrical Resistivity Tomography (ERT) is used to identify differences in root system functioning of a panel of plant genotypes by monitoring the impact of each genotype on soil water depletion patterns. We start the growing season with well-watered soil all along the ERT transect. The electrical resistivity of the soil increases (or the conductivity decreases) where water disappears through root water uptake, amongst others.  In the electrical resistivity profile this appears as a series of root water uptake bulbs or as a drying front along the transect if the plant rows are close to each other. Upon a rainfall or irrigation event, new water infiltrates in the profile, affecting the drying patterns. The effect of root water uptake is therefore most visible over a dry period. The effect of root-induced soil moisture changes is generally much larger than other, direct effects of the growing root biomass \citep{Ehosioke2020Sensing}.' |
| Figure 2. Does panel b show the electrode numbering? May be good to emphasize in the caption. | Done. |
| Line 134. I would like to have some more information about what it means to have electrode groups of 32. Does this mean that voltage and current measurements must be within the same group of 32, or can all electrode combinations be measured? | No, it's actually the number of electrodes that is connected to one plug (of 32 pins). The group has no physical constraint associated to it, we can inject within a group or between two groups. As this is confusing to talk about group of electrodes, we've removed this part of the sentence. |

| | |
|---|---|
| Line 139. Perhaps it would nice to also include the geometric factor in the table? I would also find it interesting to reflect a bit more on the selected data acquisition procedure. Would it be possible to have cross-line injections? They may have a better signal-to-noise ratio than some of the other electrode configurations used. | The geometric factor (K [m]) has been added to the appendix. Yes it would be possible to have cross-line injection (so A and M in surface cable and B and N in buried cable for example). As proof of concept, and within the time limit imposed by the one channel system and our survey frequency, we choose to select more conventional configuration. |
| Line 143. Perhaps to imprecise? I am not sure whether you are interested in the measurements. Is this not about the indicators and what we can achieve with this type of field phenotyping? | Yes, we agree this is just a side-investigation we did to test the methodology for the future, but as it is very clear from the figure 1 and 2 that something different was done here for the ERT fields than in the others, we just wanted to explain what happened as a minimum. We therefore would like to keep this, but indeed not put more emphasis on it, since it is not really crucial for the rest of the paper. |
| Line 143-148. Can this not be squeezed into an appropriate section of the results? | See comment above: since we did not develop this part further, we just wanted to explain why there are 2 different orientations. We therefore would like to keep it here. |
| Line 150. The surface is an area. This should be inserted in the soil... | Done. |
| Line 158. Does this mean that you are able to assess whether there were differences in ET0 due to the shelter? May be interesting to report this too. | We report this in figure 3 panel a with the black line, but as the precipitation deficit. We believe this is enough for this paper, but indeed, more details on shelter effects are present in the meteorological data we share together with this article. |
| Figure 3. I would say that this overview figure is better placed at the start of the presentation of the results. | That could indeed also be an appropriate place. However, since we are explaining these different steps in the methods section, we thought the scheme would be informative there already. |
| Line 180. How are the contact resistances measured or estimated? Is this an additional measurement? Please clarify. | The Subsurface Insights system is a constant voltage system (even though from an electronics perspective it could be run as a constant current system with some modest software changes). The applied voltage as well as multiple other parameters (such as number of stacks, injection length, measurement length) can be set by the user within the limits of the system.

The contact resistance is calculated from the same measurement data that is taken for resistivity. It is given for an electrode pair by dividing the voltage applied across the electrodes by the current, this is then divided by two to determine approximate resistance per electrode.

We added this to the text. |

| | |
|---|---|
| Line 186. Negative apparent resistivities were not only due to negative geometric factors? In cross-borehole studies, negative values are possible, so perhaps this is also the case for your set-up. It is a new approach, so some more reflections would perhaps be nice here. | That is correct.
Given the errors present in the "trans" quadrupoles (with A and B on surface line and M and N buried), we discarded these quadrupoles in our analysis. We also checked which quadrupoles were associated with negative apparent resistivity and it was mostly quadrupoles on the surface cable and not associated with the buried cable.  We agree that this filtering should be amended in the future if the buried array is used. |
| Line 199. Given the plea in the introduction, please make sure that all data processing are at least mentioned here and briefly described. Details can then be taken from the JUPYTER notebook. | This sentence has been rephrased as the processing steps are well described in the manuscript, only their implementation is given the notebook. There are no hidden processing steps in the notebook that are not described in the paper. |
| Line 195-199. I miss some general statements about the data quality here. Or will this topic come back later? How do your results compare to previous work? I believe most studies reported a linear relationship between reciprocal error and measured resistance? A temporal presentation of the development of the error model parameters would also be nice here – I would hope that they are rather stable in time. | In general, the data quality was good, once we had identified issues linked to the "trans" quadrupoles and the buried cable. More information on data quality is given in Figure 6 which shows the evolution of the reciprocal error in time. The error model and the evolution through time is interesting to look at and we've actually produced this figure which shows relative stability in the parameters a2 (exponent) and slight increase in the a1 parameters (slope) as the reciprocal error increases with the drought (like in fig6). We also thought of using a single error model for all timesteps to have a single relationship, however, in this study at least, we did not find that this improved the inversion compared to an error model fitted for each timestep. Further research on this might be needed.

[Figure]
 |
| Line 211. How was convergence determined? Was it possible to achieve a normalized error of 1 without accounting for additional modelling errors? | The inversion code (R2 withing ResIPy) tries to minimize a weighted RMS misfit (weighted by the reciprocal error from the fitted error model). When this weighted RMS misfit reaches 1.00, the distribution of the misfit should follow a chi^2 distribution which shows statistical convergence. The programs determine itself when to stop an iteration (i.e. when the next iteration would overfit the data: RMS misfit < 1). Appendix C provides the evolution of the inversion quality indicator with time.

The effect of modelling error was quite small in comparison to reciprocal error and did not influence the results of the inversion significantly. We neglected the modelling errors in this study for simplicity. However, going to a 3D survey with cross transects readings or including the buried electrodes, |

| | would have certainly led to larger modelling error that should be included in addition to the reciprocal error. |
|---|---|
| Line 220. Integrate equation in the text. | Done |
| Line 224. I find this a little bit short. Some more reflection on this important calibration seems justified to me. The scatter is considerable. How does this affect the presented results? | We kept this short, since we just want to illustrate the data and what can be done with it. But given the scatter and the range of the data which should still be expanded, we did not use the relationship further on in the processing pipeline. At the moment all indicators are calculated on EC and deltaEC data and not estimated soil moisture.

We added the text below to this paragraph to clarify this: 'The data largely follows the expected trend, but also exhibits significant scatter. Since we want to investigate the pedophysical relationship further in future experiments, we have not used it in the current manuscript to compute indicators based on estimated soil moisture yet. ' |
| Line 231. I think the equation should also be provided in the text (or only in the text). | Done, but also kept it in the overview table. |
| Figure 9. I realize that a lot depends on the visualization here, but the coverage of the different electrode configurations does not seem high in this one. Perhaps the cumulative sensitivity or resolution should additionally be presented to provide some evidence that profiles can adequately be obtained? | That's a good point.
There are multiple ways to assess the coverage (sensitivity, depth of investigation, resolution matrix…). Since we have investigated the experimental layout in an earlier contribution (EAGE NSG extended abstract) and although it is always an important thing to check, we prefer not to include a plot and details on this in the article this time. However, the availability of the code and entire data set to the readers will allow them to play with the different metrics.

However, two things allow us to interpret profiles up to 1 m max: (1) close to constant sensitivity during the season and (2) depth of plant induced signal in the inversion.
For completeness, we checked the sensitivity pattern of one transect taken at a wet (June) and dry (August) moment and the sensitivity pattern stays similar with most sensitivity concentrated in the first meter (according to a 'subjective' threshold, fig below). Secondly, the inverted transect shows well similar changes to the apparent values at depth up to 1 m maximum, hence why we choose to exploit the profiles up to 1 m depth. Note that this will depend a lot on the sequence used and the plant we target as the depth of their root system will be different. |

[Figure]

| | |
|---|---|
| Line 255-260. Would be good to discuss whether the observed differences match the expectations for the genotypes. | Unfortunately, little to no information is available on the belowground behavior of our genotypes. We do have some experience and data on aboveground behavior of the genotypes from earlier studies.

We state higher in the text that the Lenka genotype is known be resistant to drought thanks to the slow canopy wilting drought trait. Pro-1 and Hermes do not have this trait, but the Pro-1 genotype has a more compact habitus that can potentially make it more drought tolerant.

We also know that both Lenka and PRO1 are preferred by the industry as they both have a higher protein content which is preferred for processing.

This of course does not say much about what is to be expected belowground…

To link both belowground and aboveground information to each other is now the subject of ongoing research and hopefully future publications. |
| Figure 10. I am confused about the selection of dates here. Do I understand correctly that all these measurements are made shortly after sowing? Can we really attribute the observed differences to plants in this case? How quick will the root system develop after sowing at 21.06? | As the Appendix A show, soybean was sown on 24/5. ERT monitoring started on 21/6. It is true that it would be good to highlight this more clearly in the text, which we did now under 'data acquisition/ERT'. 'For the POC experiment, we have sown the three soybean genotypes on 2023-05-2023 (more information on timing of different agronomic and experimental events, see Appendix A). '

Extra information to answer the questions:
These are the same dates as in the pseudo sections in Figure 9. 23/6 is close to the vegetative stage 4: presence of the 5 leaf (or the 4th trifoliate leaf) and therefore already some transpiration. The last date is close to reproductive stage 2, a moment in which already 50% canopy cover is observed. We have no early measurements of the root system itself, but given these plant developments, it can be assumed that the root system is following. Exactly what we are showing with the ERT.
We will write another manuscript on combining all data of the trial and showing more details from the UAV etc, but in the figure below you can see that the plant was clearly developing |

at these moments. Around the last date we show, the plants are already flowering.

[Figure]

Overview of crop growth stages, environmental conditions, and phenotyping data during the POC2023 experiment. (a) key crop phenological stages of soybean from sow to, progressing through the vegetative stages, Vv, and the reproductive stages, Rr. (b) Cumulative Water Deficit (CWD) and the daily rainfall. Irrigation = bright blue bar. (c) Change in Volumetric Water Content (ΔVWC) compared to sowing date (depth = 0-30 cm). (d) Canopy Height (CH) and (e) Canopy Cover (CC) from RGB camera. (f) Soil-Adjusted Vegetation Index (SAVI) from multispectral camera (g) Temperature Difference to Reference Evapotranspiration (ΔT/ETo) from thermal camera. (h) Apparent Electrical Conductivity ($\rho_a$) at a pseudodepth of 30 cm and (i) change in Apparent Electrical Conductivity normalized by initial conductivity ($\Delta Eca/Eca_0$):, with error bars indicating standard deviation. The drone and ERT sections show the average of all 3 genotypes within their treatment.

| | |
|---|---|
| Line 275. Reference to figure missing near the start of the paragraph. | Added |
| Line 315. Not sure it was not designed for that. Perhaps state that you took it considerably | Ok. We removed this sentence and added at the end of the paragraph: 'With HYDRAS, we take belowground field |

| | |
|---|---|
| further by successfully deriving indicators suggesting significant differences between genotypes. | phenotyping one step further with the fully automated pipeline, derived phenotyping indicators and a field setup for breeding trials including rainout shelters. ' |
| Line 334. Instead of inverting for the electrode locations, you could perhaps just optimize the shift? This would significantly reduce the degrees of freedom. I also think that you did not really provide proof that the deeper cable is helping with resolution. I can buy it, but perhaps there is scope to address this too in the manuscript. I did not find it excessively long at the moment... | We did show this in another earlier contribution on the design of ERT in HYDRAS: https://doi.org/10.3997/2214-4609.202120221. We prefer not to address this point in this article, if that is ok.

This is a figure from that extended abstract:

[Figure]
 |